# Vertical redistribution of salt and layered changes in global ocean salinity

Chao Liu[1], Xinfeng Liang [1,6], Rui M. Ponte[2], Nadya Vinogradova[3,4] & Ou Wang[5]

Salinity is an essential proxy for estimating the global net freshwater input into the ocean. Due to the limited spatial and temporal coverage of the existing salinity measurements, previous studies of global salinity changes focused mostly on the surface and upper oceans. Here, we examine global ocean salinity changes and ocean vertical salt fluxes over the full depth in a dynamically consistent and data-constrained ocean state estimate. The changes of the horizontally averaged salinity display a vertically layered structure, consistent with the profiles of the ocean vertical salt fluxes. For salinity changes in the relatively well-observed upper ocean, the contribution of vertical exchange of salt can be on the same order of the net surface freshwater input. The vertical redistribution of salt thus should be considered in inferring changes in global ocean salinity and the hydrological cycle from the surface and upper ocean measurements.

[1] College of Marine Science, University of South Florida, St Petersburg, FL 33701, USA. [2] Atmospheric and Environmental Research, Lexington, MA 02421, USA. [3] NASA Headquarters, Science Mission Directorate, Washington, DC 20546, USA. [4] Cambridge Climate Institute, Somerville, MA 02145, USA. [5] Jet Propulsion Laboratory, Pasadena, CA 91109, USA. [6] Present address: School of Marine Science and Policy, College of Earth, Ocean, and Environment, University of Delaware, 700 Pilottown Road, Lewes, DE 19958, USA. Correspondence and requests for materials should be addressed to X.L. (email: xfliang@udel.edu)

Over the past decades, the warming climate has resulted in the melting of glaciers and ice sheets[1] and consequently, the ocean received net freshwater. Besides thermal expansion, the net freshwater input is a major contributor to global sea-level rise[2]. Because of the net freshwater input, the global ocean likely displayed a freshening tendency[3,4]. Ocean salinity could, therefore, be useful in estimating the global net freshwater input to the ocean and understanding sea level changes[5,6]. In a broader context, ocean salinity is recognized as an essential climate and ocean variable of the Global Climate Observing System[7] with a growing number of efforts being dedicated to interpreting the changes in ocean salinity and its linkages with the global freshwater cycle[8–15].

The expanding in situ salinity measurements, which are obtained mainly through the Argo program, have been utilized to improve our understanding of ocean salinity and the surface freshwater flux[16,17]. Although these Argo-based estimations of surface freshwater flux consider horizontal advection and time tendency contributions, the salt flux at the bottom of the mixed layer is mostly ignored[17]. As many studies have revealed[10,18,19], the vertical redistribution of salt related to ocean advection and mixing processes is critical in controlling salinity variability in the mixed layer and helps determine the changes in freshwater flux using salinity data.

To understand salinity and freshwater dynamics, one needs to reliably represent ocean salinity and the related ocean dynamical processes globally and over the full depth. Unfortunately, at present, global-scale salinity measurements are limited to the upper ocean and only available after the Argo program started[1]. Alternatively, ocean synthesis, involving the combination of ocean models, atmospheric forcing, and ocean observations, could potentially provide more comprehensive and physically complete information about ocean changes[20,21]. In addition to estimates of salinity over the global ocean and its full-depth, ocean synthesis provides a number of unobservable quantities (e.g., vertical velocity) and can be used to quantify the various dynamical terms that contribute to ocean salinity changes.

Here, we examine changes in global ocean salinity, temporal means of vertical salt flux, and other terms that determine the global ocean salinity budget using an available ECCO (Estimating the Circulation and Climate of the Ocean) state estimate[22–24]. In the ECCO estimates, the ocean heat and salt contents are conserved, and an accurate budget analysis can be conducted[19,25–27]. In this study, we will focus on changes of horizontally averaged salinity in different layers of the global ocean and explore the role of the vertical redistribution of salt in determining the salinity changes in the well-observed upper ocean (hereinafter defined as the layer above 2000 m.) In addition, implications for interpreting the changes in the upper ocean salinity and inferring information about the total freshwater input at the surface from existing observations will be briefly discussed.

## Results

**Changes in Global Ocean Salinity.** The temporal variation of the horizontally averaged salinity anomaly shows a layered structure (Fig. 1). Near the surface (above 200 m), the salinity anomaly

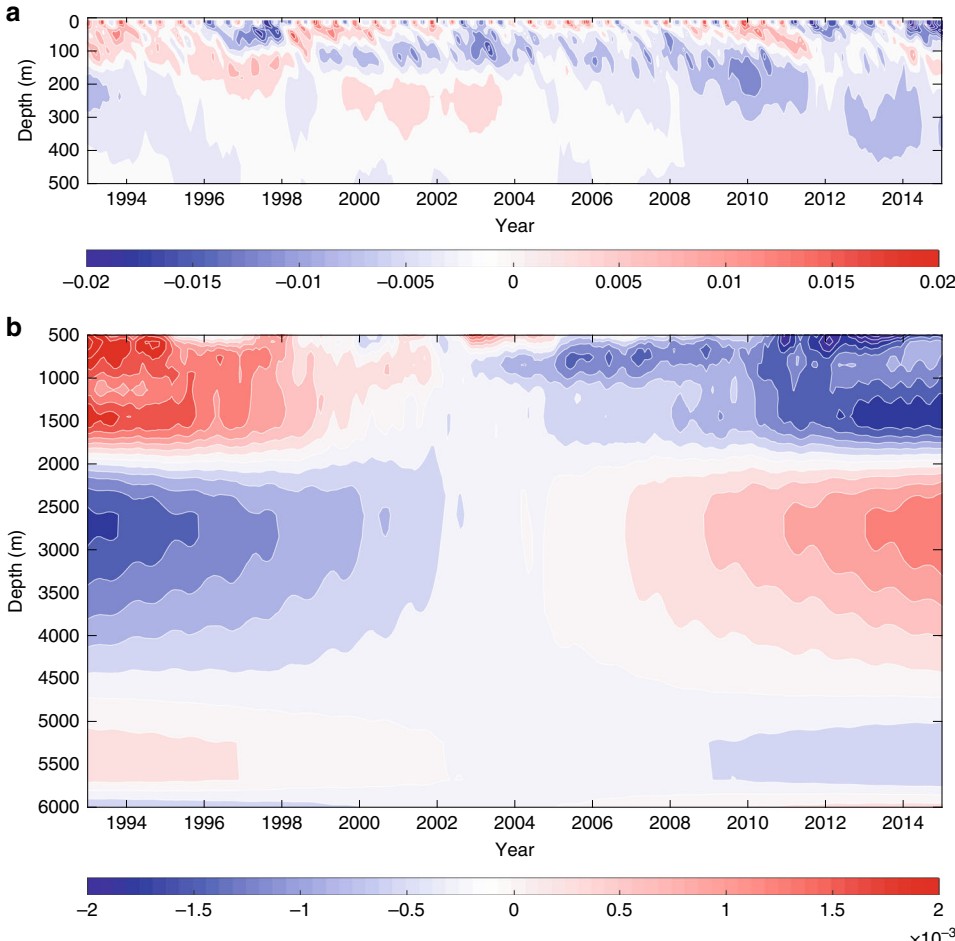

**Fig. 1** Time evolution of the globally averaged ocean salinity. Salinity anomalies (unit: psu) are calculated as deviations from the temporal means over the period January 1993 to December 2014. Salinity anomaly for the upper 700 m (**a**) and for the layer 700–2000 m (**b**). Note that the range of salinity anomaly varies in each panel. The high-frequency signals are residuals after removing the fitted annual variability

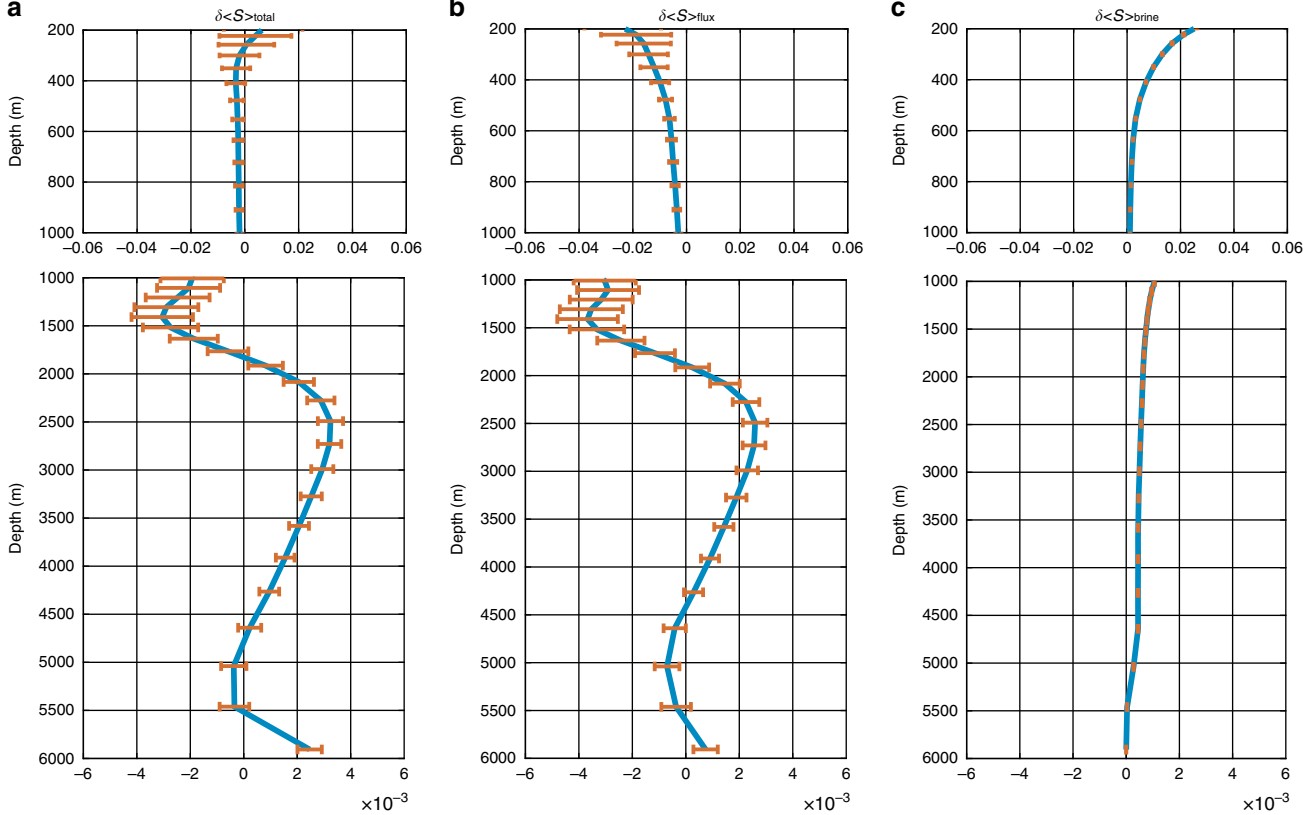

**Fig. 2** 22-year changes of the globally averaged ocean salinity. **a** Changes of the horizontally averaged ocean salinity ($\delta<S>_{total}$, unit: psu) calculated as December 2014 minus January 1993 values. **b** Salinity changes due to the net vertical salt flux ($\delta<S>_{flux}$, unit: psu). **c** Salinity changes due to the brine rejection ($\delta<S>_{brine}$, unit: psu). The upper 200 m are excluded to avoid direct impacts from the intense high-frequency variability. The error bars are calculated as $\sigma/\sqrt{n}$, where $\sigma$ is the temporal standard deviation of the corresponding term, and $n$ is the degrees of freedom (261 in this study)

displays high-frequency variability along with long-term freshening. Below 500 m, weak high-frequency variability is superposed on strong long-term changes, which reverse at about 2000 and 4500 m. Specifically, the layer between 500 and 2000 m as well as the layer below 4500 m show freshening, but the layer between 2000 and 4500 m displays a salinification. Freshening in the upper and abyssal oceans has been either described or suggested in previous studies[28–31]. However, as far as we are aware, a salinification in the deep ocean between about 2000 and 4500 m on the global scale has not been reported before and should be examined further in the future. Note that below 500 m the salinity anomaly changed signs around 2005, the year when the Argo program reached global coverage. However, such change in sign mainly reflects the presence of a linear trend, which naturally yields a change in sign around the middle of the examined period, rather than possible impacts of assimilating Argo measurements.

Changes in the horizontally averaged salinity in the ocean interior can be attributed to the divergence/convergence of the net vertical salt flux due to advection and diffusion processes and the flux associated with the vertical redistribution of salt rejected during the formation of sea ice, which is the subject of a separate parameterization in ECCO estimate[23] and in this paper termed brine rejection. (See Methods for more details.) Fig. 2 displays the profile of the 22-year (Jan. 1993—Dec. 2014) changes in the horizontally averaged salinity ($\delta<S>_{total}$), as well as the separate parts due to the net vertical salt flux ($\delta<S>_{flux}$) and the parameterized brine rejection ($\delta<S>_{brine}$), where $\delta<S>_{total} = \delta<S>_{flux} + \delta<S>_{brine}$. It is clear that the layered structure of $\delta<S>_{total}$ in the vertical direction is determined by the part associated with the vertical salt flux, $\delta<S>_{flux}$. In particular, the

contribution of $\delta<S>_{flux}$ to $\delta<S>_{total}$ (hereinafter calculated as $|\delta<S>_{flux}|/(|\delta<S>_{flux}| + |\delta<S>_{brine}|)$) below 600 m is always over 70%. The contribution of $\delta<S>_{brine}$ to the total salinity change decreases with depth and is mostly limited to the upper 600 m.

The spatial patterns of temporal means of the net surface freshwater flux and the surface salt flux, as well as the bidecadal changes of ocean salinity averaged within different layers of the global ocean, are presented in Fig. 3. Note that the net surface salt flux is due to the net exchange between the ocean and sea ice over the examined period. As expected and suggested by previous studies[19,26], the discrepancies between the spatial patterns of surface fluxes and changes of salinity in Fig. 3 demonstrate the importance of the ocean dynamics in setting the spatial patterns of the upper-ocean salinity changes.

The bidecadal salinity changes for various vertical layers (Fig. 3c–f) show significant regional patterns and vertical structures. For the top-to-bottom averaged salinity, the most significant changes appear at high latitudes, such as the Arctic and the subpolar North Atlantic Ocean, as well as in coastal regions, such as the west coast of Australia. For the upper 500 m, which contributes the most to the changes of the whole water column (Fig. 3c, d), the spatial distribution of the bidecadal salinity change is generally consistent with previous studies[10,26], such as a salinification in the North Atlantic Ocean and the western tropical Pacific Ocean. For the layer 500–2000 m, freshening becomes dominant over most of the global ocean, particularly in the tropical and subtropical Indian and Pacific Oceans. For the layer below 2000 m, small positive change dominates the entire Pacific Ocean and the Southern Ocean while small negative patches appear in the northern part of the Indian

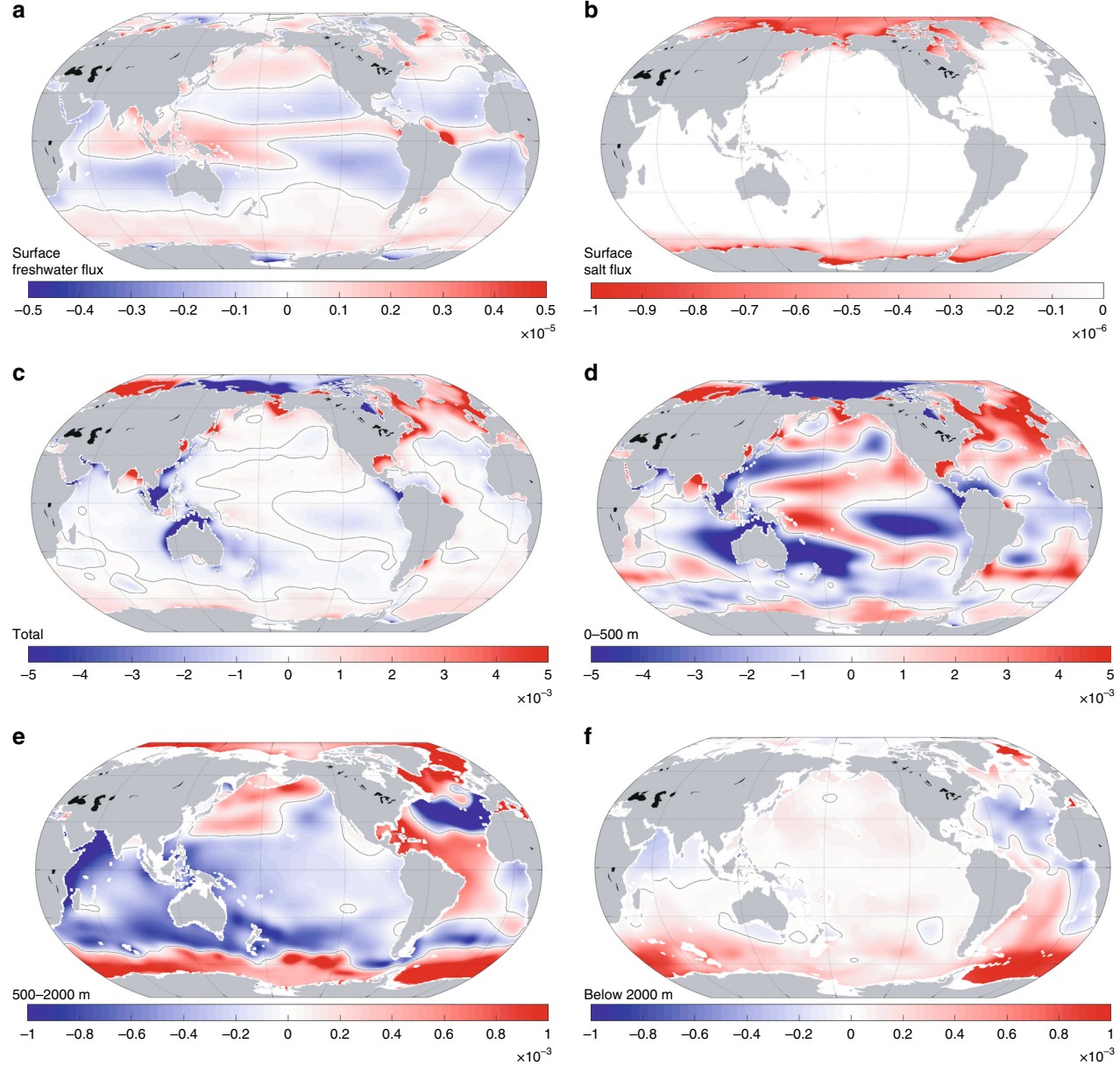

**Fig. 3** Surface fluxes and trends of salinity in sampled layers. **a** 22-year mean of the ocean surface freshwater flux (converted to psu m s$^{-1}$). **b** Net surface salt flux into the ocean due to sea ice formation processes (psu m s$^{-1}$). In **a** and **b**, positive values stand for net downward (upward) freshwater (salt) transport. **c–f** 22-year trends (psu year$^{-1}$) of ocean salinity averaged over the whole water column (**c**) and three different layers (**d–f**). Note that the scale varies in different panels

Ocean and portions of the Atlantic Ocean. The most significant positive values in this layer appear in the Southern Ocean and could probably be related to the changes in the depth of isopycnal (heaving) and/or the volume reduction in the bottom waters[32].

**Global Ocean vertical salt fluxes**. Due to its dominant role in producing the vertically layered structure of global salinity changes (Figs. 1 and 2), here we directly examine the vertical salt flux from ECCO (Fig. 4). The 22-year and horizontally averaged vertical salt flux, $<\bar{F}>_{net}$, shows a net upward salt transport that decreases with depth in the upper 700 m over the examined period. For instance, at 200 m there is an upward salt flux of $0.84 \pm 0.56 \ 10^{-8}$ psu m s$^{-1}$, equivalent to $2626 \pm 1751$ Gt year$^{-1}$ of freshwater input into the global ocean, which decreases to near zero at 700 m. Below 700 m, the direction of $<\bar{F}>_{net}$ changes from upward to downward, but its positive gradient remains until

about 2000 m, suggesting divergence of vertical salt flux for that layer. Consequently, a freshening appears between 700 and 2000 m. Below 2000 m, $<\bar{F}>_{net}$ is downward until about 4000 m and shows a negative gradient until about 4500 m, consistent with the salinification in the layer 2000–4500 m (Fig. 2).

To explore the roles of different dynamical processes in the vertical redistribution of salt content, we further separate the net vertical salt flux ($<\bar{F}>_{net}$) into advective ($<\bar{F}>_{adv}$) and diffusive ($<\bar{F}>_{diff}$) terms (Fig. 4b, c). $<\bar{F}>_{adv}$ consists of contributions from the large-scale ocean circulation and parameterized mesoscale eddies; $<\bar{F}>_{diff}$ includes contributions of both isopycnal and diapycnal mixing (see Methods section for more information). Despite the discrepancy in uncertainties that are associated with temporal variability, the contribution of $<\bar{F}>_{adv}$ to the net vertical salt flux is comparable to that of $<\bar{F}>_{diff}$. Generally, the horizontally averaged advective vertical salt flux, $<\bar{F}>_{adv}$, is

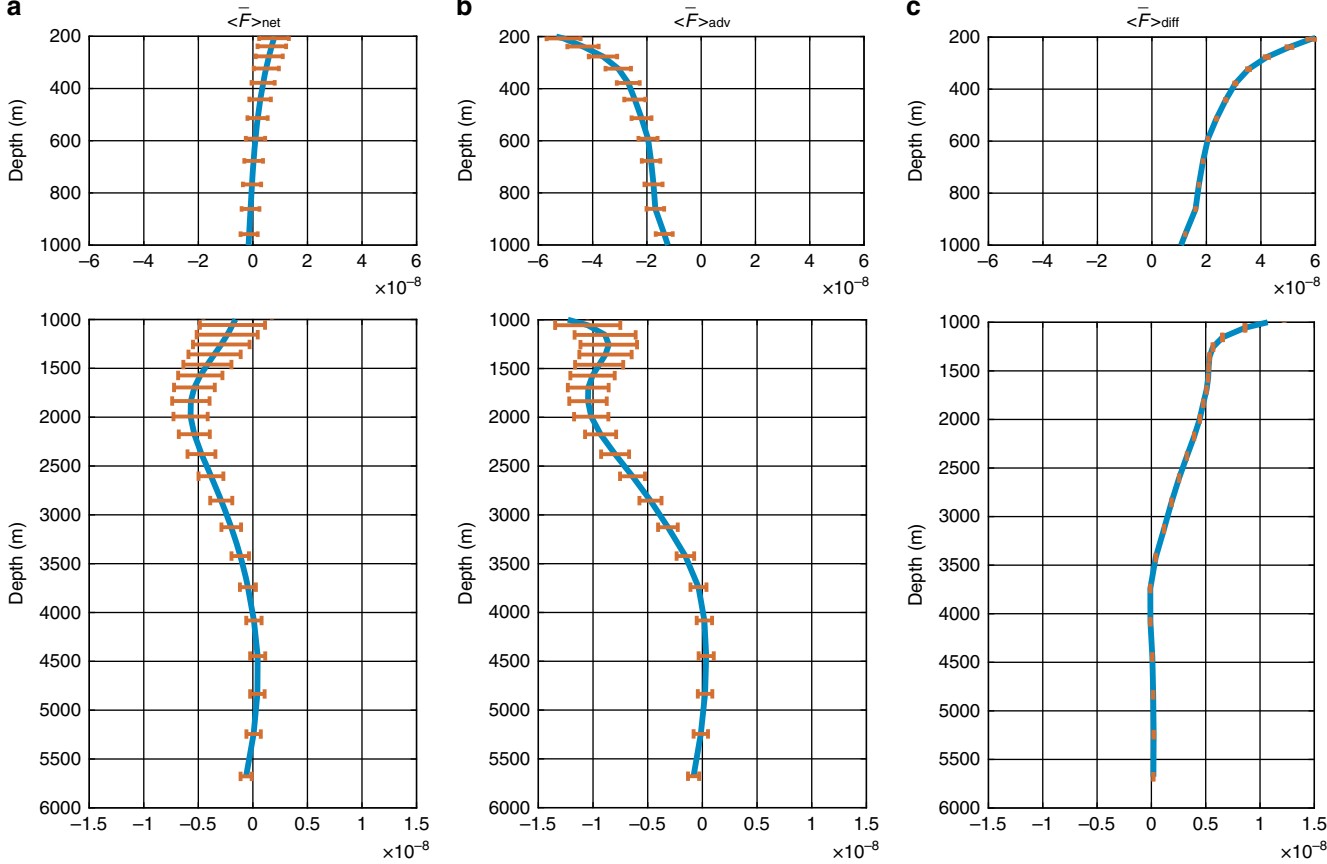

**Fig. 4** 22-year means of the globally averaged vertical salt fluxes. **a** Temporal means of the horizontally averaged net vertical salt flux from January 1993 to December 2014 ($<\bar{F}>_{net}$, unit: psu m s$^{-1}$). **b**, **c** same as in (**a**) but for the advective ($<\bar{F}>_{adv}$, **b**) and diffusive ($<\bar{F}>_{diff}$, **c**) vertical salt fluxes. Note that $F_{net}$ is the sum of $F_{adv}$ and $F_{diff}$. In all panels, positive values stand for upward salt flux. The upper 200 m are excluded to avoid direct impacts from the intense high-frequency variability. The error bars are calculated as $\sigma/\sqrt{n}$, where $\sigma$ is the temporal standard deviation of the corresponding flux term, and $n$ is the degrees of freedom (261 in this study)

negative, implying the circulation and eddies transport salt downward in most of the examined depth range. In contrast, the diffusive counterpart, $<\bar{F}>_{diff}$, is positive at all depths. The upward diffusive salt flux mainly occurs in the high-latitude North Atlantic and the Southern Ocean (Fig. 5e, f), where the isopycnal mixing dominates the vertical diffusive fluxes[33].

To identify the critical regions for the vertical redistribution of salt, we also examine the spatial patterns of $<\bar{F}>_{net}$, $<\bar{F}>_{adv}$ and $<\bar{F}>_{diff}$ at two sample depths (500 and 2000m). For the net vertical salt flux at 500m (Fig. 5a), relatively strong salt exchanges of both signs appear on various spatial scales over the global ocean. In particular, the Southern Ocean, western boundaries of major ocean basins, and the North Atlantic show evident relatively small-scale features. In the deep and abyssal oceans (Fig. 5b), the vertical salt exchange is generally weak, and the strongest vertical salt exchange appears in the Southern Ocean and the high latitude North Atlantic, confirming the essential role of these regions in the vertical transport of ocean properties and tracers[25,34,35]. Also, the spatial patterns of the vertical salt flux are primarily determined by the advection, i.e., vertical velocity, which is closely related to wind stress near the surface and the large-scale bathymetry in the deep and abyssal oceans[36]. Note that although the diffusive term is comparable to the advective term when integrated globally (Fig. 4), it is much smaller regionally except in the high latitude regions where isopycnal mixing becomes vital for vertical diffusive flux (Fig. 5).

## Discussion

Because the existing salinity measurements are mostly limited to the surface and upper ocean, the ECCO estimates of salinity in the deep and abyssal oceans are mainly constrained by repeated hydrographic measurements, ocean measurements that represent the vertically integrated effects of salinity and temperature changes (e.g., sea level anomalies from altimeters), and ocean dynamics represented in the MITgcm. Uncertainties in the ECCO salinity estimates are expected, especially in the deep and abyssal oceans where direct hydrographic data constraints are mostly missing. Also, because of the coarse resolution, impacts of mesoscale eddies are parameterized[37] rather than explicitly resolved. Nevertheless, the estimates are generally consistent with previous regional studies on deep and abyssal ocean salinity changes and provide a unified picture of the global salinity changes. A comparison between the salinity estimates from ECCO and Argo over the overlapping period is displayed in Supplementary Figs 1 and 2. Patterns revealed in salinity estimates from ECCO are also compared with previous studies and presented in Supplementary Note 1.

Our quantification of the global ocean vertical salt flux and its contribution to ocean salinity changes in different layers has important implications. In particular, net upward salt fluxes above 700 m were revealed (Fig. 4). The vertical redistribution of salt in the upper ocean could significantly compensate for the salinity changes induced by the net freshwater flux into the ocean.

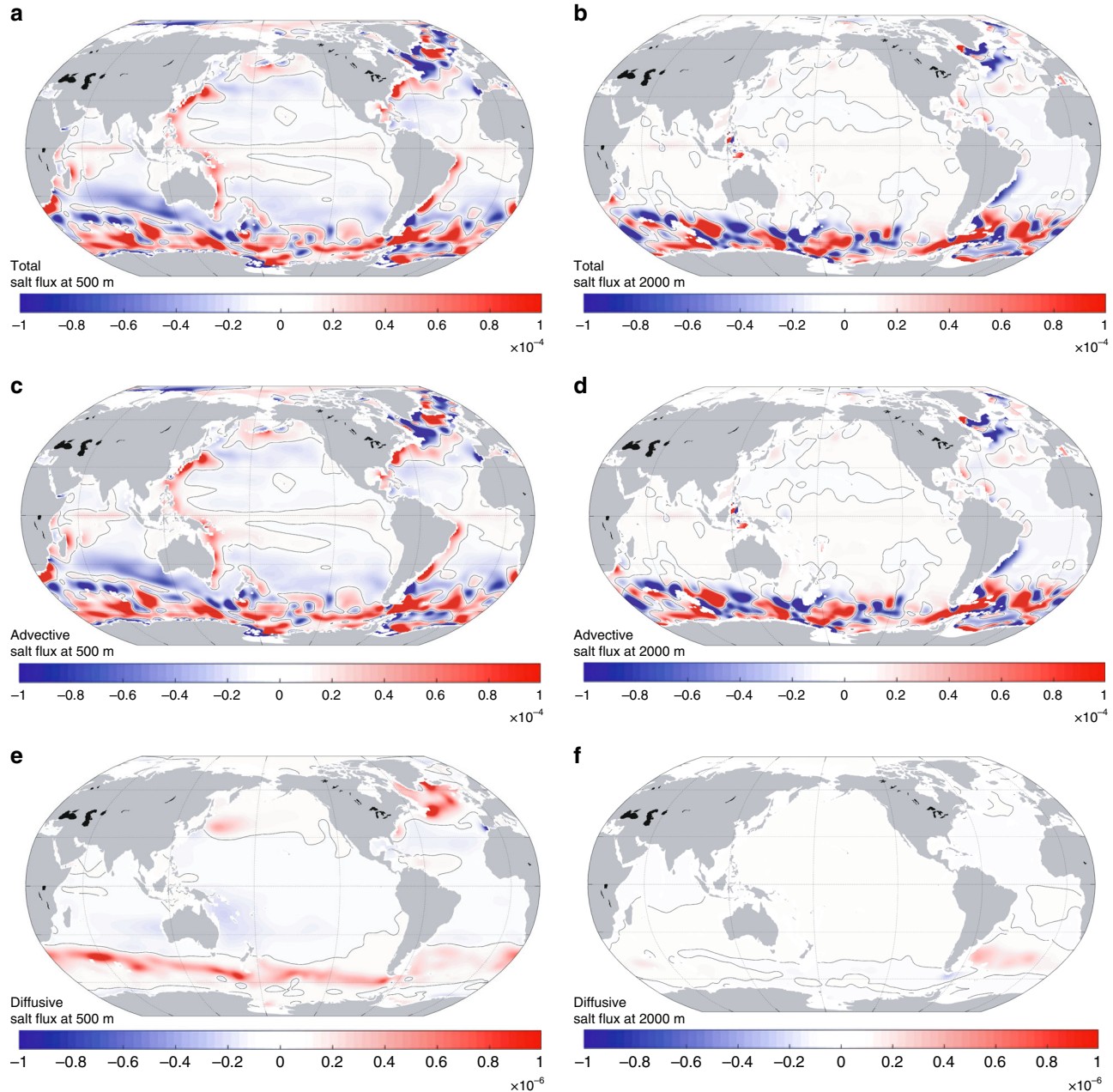

**Fig. 5** Spatial distribution of the ocean vertical salt fluxes. **a**, **c**, **e**, 22-year means of the net (**a**), advective (**c**), and diffusive (**e**) vertical salt fluxes at 500 m. **b**, **d**, **f**, 22-year means of the net (**b**), advective (**d**), and diffusive (**f**) vertical salt fluxes at 2000 m. In all panels, positive values stand for upward salt transport psu m s$^{-1}$. Note the scale varies in different panels

For instance, the globally averaged net surface freshwater flux into the ocean from ECCO is 664 Gt year$^{-1}$, corresponding to a vertical salt flux of $0.20 \times 10^{-8}$ psu m s$^{-1}$, which is smaller but of the same order of magnitude as the estimated vertical salt flux at 200 m ($0.84 \times 10^{-8}$ psu m s$^{-1}$) and is close to the rate of land ice loss from glaciers and ice sheets. Thus, in inferring the surface freshwater input from the relatively well-observed surface or upper ocean salinity changes, one should consider the vertical exchange of salt inside the ocean.

One way to avoid the impacts of the vertical salt redistribution in utilizing the available ocean salinity measurements is to use the portion above the turning depth of the global vertical salt flux. Our estimate implies this depth could be around 700 m, which is within the range of Argo observations and we have good coverage of the global ocean to that depth. The more accurate the turning

depth we can get, the less bias the estimated surface freshwater input will have. However, considering the potential uncertainties in the current salinity estimates, more studies are needed to examine this critical turning depth further to reach robust conclusions.

In reality, the global net freshwater input includes the contributions of melting land ice and sea ice, the latter of which does not affect sea level, and of changing evaporation and precipitation. If the global net freshwater input can be accurately inferred from salinity measurements, closure of the freshwater budget over the global ocean is potentially achievable by utilizing multi-sources of observations—e.g., Argo salinity measurements, land ice measurements from GRACE[38], air-sea freshwater fluxes estimates[39], sea surface height altimeter data, although very likely including substantial uncertainties. By examining the spatial

patterns of ocean salinity changes and freshwater input near the surface, one can potentially infer redistribution in freshwater from rivers, meltwater, rain over the global oceans as well.

Furthermore, like ocean salt content, the ocean heat content change is strongly affected by the vertical exchange processes as well[25]. So, to more accurately interpret changes in the upper ocean or to infer information about climate change from the upper ocean observations, we need to better understand the vertical exchanges of heat and salt between upper and deep oceans, as well as the dynamical processes that are involved, such as vertical velocity, and isopycnal and diapycnal mixing. Regions where the most robust vertical exchanges appear, particularly the Southern Ocean and the North Atlantic, should receive high priority in developing future observations. The ongoing global Deep Argo program[40,41], once it reaches global coverage, will significantly advance our understanding of the deep ocean changes as well as the exchanges between upper and deep oceans.

## Methods
**Data**. This study is based on the latest release of the ECCO estimate (labeled v4r3 for Version 4 Release 3)[23,24,42]. The ECCO estimate can be interpreted as a least-squares fitting of the MIT general circulation model (MITgcm) to a large volume of satellite and in situ measurements. In contrast to other ocean synthesis products, the ECCO estimate does not introduce artificial jumps during data assimilation, which makes it both dynamically consistent and close to the available observations within the specified levels of data uncertainty[22]. Also, ocean properties (i.e., temperature and salinity) and mass (volume) are conserved to machine precision[23]. These unique features of the ECCO estimate allow accurate budget analyses for ocean heat and salinity changes[19,25–27]. The ECCO v4r3 solution provides monthly averaged estimates from Jan. 1992 to Dec. 2015 (total of 288 months) on a vertical grid with 50 layers of various thickness. The horizontal resolution of the estimates varies between 1/3° near the equator and 1° in the midlatitudes. For more information about ECCO v4r3, refer to previous studies[23,24].

**Changes in Global Ocean Salinity**. The ECCO salinity estimates are horizontally averaged and examined at various depths. In the ocean interior, changes in horizontally averaged salinity are solely due to the vertical redistribution of salt. In addition to the net vertical salt flux associated with common advective and diffusive processes, salt rejected into the ocean during the formation of sea ice is redistributed vertically. In ECCO v4r3, the redistribution of the salt associated with brine rejection is parameterized[43,44] and acts as a source for the salinity budget in the subsurface ocean. Although the brine rejection term can affect the ECCO salinity estimates in the ocean interior[23], it is only significant at high latitudes and is always of the same sign in the whole water column. In this study, the focus is on the vertical salt fluxes related to vertical advection and diffusion, which determine the vertical structures of horizontally averaged salinity.

**Surface freshwater flux and surface salt flux**. In ECCO v4r3, the surface freshwater flux consists of contributions of open ocean evaporation, precipitation, and river runoff. Evaporation is calculated conventionally with the bulk formulae[45]; the initial guess of precipitation is from the ERA-Interim re-analysis[46]; and the runoff is a seasonal climatology estimate[47]. Note that in ECCO v4r3, the glacial runoff is not explicitly specified in the model configuration. Since many of the near-surface atmospheric variables, including temperature and precipitation, are control variables in ECCO v4r3, they were adjusted in the optimization to make the estimates better fit the observations[23]. Therefore, impacts of the terms and processes that are not explicitly prescribed, such as the glacial runoff, can be incorporated through the adjustment process. The surface salt flux only appears where sea ice is present. In ECCO v4r3, sea ice carries a salinity of 4 psu. When sea ice forms or melts, there is a net salt exchange between ocean and sea ice. For more details about the surface boundary conditions, refer to Forget et al.[23].

**Oceanic vertical salt flux**. The net vertical salt flux ($F_{net}$) in ECCO v4r3 consists of advective ($F_{adv}$) and diffusive ($F_{diff}$) terms. Moreover, the vertical advective salt flux $F_{adv}$ is the sum of the Eulerian-mean transport $wS$ and the parameterized eddy induced transport, also called bolus transport, $w^*S$. Here $S$ is the ocean salinity; $w$ is the vertical velocity associated with large-scale circulation; and $w^*$ is the bolus velocity, representing the advective impacts of mesoscale eddies[37]. The diffusive term $F_{diff}$ includes contributions of diapycnal mixing, convective adjustment, and isopycnal mixing. The reason that isopycnal mixing also contributes to the vertical diffusive salt flux is due to the existence of salinity gradient along isopycnals. As revealed in previous studies[25], isopycnal mixing is also important for the vertical heat transport, particularly in regions with steep isopycnal slopes. In ECCO v4r3 the background diapycnal mixing and isopycnal mixing were control parameters

and were adjusted from spatial uniform values to spatial varying estimates, as part of fitting the model to the observations[23,48].

**Error bars**. In this study, we focus on interpreting the bidecadal changes in ocean salinity. As such, for both salinity and vertical salt fluxes, sinusoidal annual cycles were fitted and removed before calculating the means and changes. The first and last years are also excluded for quality reasons. The uncertainties of these variables are defined as $\sigma/\sqrt{n}$ where $\sigma$ is the temporal standard deviation, and $n$ is the degrees of freedom (261 after annual cycle removal for 22-year monthly data). Note that the uncertainties presented in this study only represent the impacts of temporal variability resolved within the ECCO v4r3 estimates. Other uncertainties associated with data errors, limited global coverage, inaccurate models, regional variability, etc. are not included. Therefore, the real uncertainties, which are difficult to quantify, can be much larger than the values presented in this study.

## Data availability
All the ECCO v4r3 data are available at the ECCO group website (https://ecco.jpl.nasa.gov/).

## Code availability
The source codes for ECCO v4r3 can be downloaded online (https://ecco.jpl.nasa.gov/); the scripts used to make the plots in this paper are available from the corresponding author on request.

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

## Acknowledgements

X.L. is grateful for the support from the National Science Foundation through Grant OCE-1736633 and the Alfred P. Sloan Foundation. C.L. was partially supported by the NASA Earth and Space Science Fellowship (NESSF) through Grant 80NSSC18K1320. Comments on early versions of the manuscript from Christopher Piecuch greatly improved this paper. The ECCO project is supported by the NASA Physical Oceanography, Cryospheric Science, and Modeling, Analysis and Prediction programs.

## Author contributions

X.L. and C.L. conceived the study. C.L. conducted the analyses. X.L. and C.L. drafted the manuscript. R.M.P., N.V., and O.W. contributed to discussing the results and revising the manuscript.

## Additional information

**Competing interests:** The authors declare no competing interests.

