## [Peer Review File · Nature Communications]

Reviewers' comments:

Reviewer #1 (Remarks to the Author):

This paper is well written and uses the ECCO model to make a consistent picture of salt redistribution in the vertical over the past two decades. It makes the point that sub-mixed layer fluxes matter when interpreting surface water fluxes. I have no problem with that; it's certainly a valid point. But so what? Having done the analysis what can you say about:

1. The mean salinity of the ocean? Is it freshening due to ice melt? It is not likely that we can yet tell but what would be needed to see that? This must depend on salt and mass conservation in the ECCO model, so a confirmation that they are indeed truly conserved is important. Some discussion of the surface boundary conditions on salinity and freshwater that are used in ECCO is appropriate. Surely the surface salt flux must be zero everywhere! (Fig. 3b). Water fluxes are non-zero due to evaporation, precipitation and ice formation but salt fluxes should always be zero.

2. What are the implications for a changing water cycle? That is, it is stated that the sub-surface flux is comparable to the surface flux, so the obvious question is how do these results affect the interpretation of Durack et al (2012) on the trends in the global water cycle. It seems that some modifications of the regional water cycle changes inferred by Durack's SSS pattern trends is called for. This should be a more tractable problem than (1) above and would show the value of the calculation. In addition to these fundamental motivating questions, the basic analysis can be challenged. That is, just describing the vertical distribution of salt mixes up trends in heaving of density surfaces with vertical mixing across density surfaces. It would be nice to see this analysis re-done in density space rather than depth to properly get at the vertical mixing. It is also troubling to see such large error bars in figures 2 and 4.

Because of these fundamental issues, I do not recommend publication in its present form. In addition I think the term "salt plumes" is rather odd and at least needs better definition. Salt plumes from ice formation would be very small-scale (cm) so way below model resolution. Do you mean something like the Mediterranean out-flow water cascading down the slope in the Atlantic?

I also take issue with the ECCO salinity maps that in some cases do not look much like the ARGO maps (Figures A1 and A2) . This is especially true of the mid-depth Atlantic in A2!

Reviewer #2 (Remarks to the Author):

Review of "Vertical Redistribution of the Global Oceanic Salt Content" by Liu et al.

Let me first share my name with the authors: William Llovel.

The paper deals with ocean salinity change on a bidecadal basis based on the last version of the ECCO ocean reanalysis (ECCOv4r3). The authors justify the need to investigate the salinity change as it is an essential proxy for detecting and estimating the global freshwater input into the ocean. In the context of the actual global warming, mountain glaciers and ice sheets are losing mass increasing the freshwater input into the oceans. Therefore, the topic of the study is a great interest. Because of the lack of in situ measurements of salinity compare to temperature change for the last decades, salinity change and freshwater flux have been under investigated compared to temperature change.

One of the fundamental results of the study is the importance of the vertical redistribution of salt while estimating the net freshwater input. According to the authors, if we ignore the vertical redistribution of salt, the freshwater budget may be biased by 42.9% at 200m, 17.2% at 500m and less than 2% at 700m on a global basis.

This manuscript is well-written, and the results are very interesting, the conclusions reached are sufficiently important to be published for future reference. The methodology is understandable, and the authors do not over-interpret the results -which is very good-. Thus, I have no major fundamental concerns about the paper.

I do, however, have a few concerns and questions that I would like the authors to address before recommending the paper for acceptance:

- ECCO ocean reanalysis is assimilating in situ and remote sensing observations. Therefore, the reanalysis tends to match the model to the existing observations. However, is this fit done for the good reason? I am not sure this is the focus of the paper but few words might be useful to warn the reader about the skills and the limitations of such a tool.

- GRACE data are assimilated in the ECCO reanalysis. But, as I understand, only GRACE data over the oceans have been considered. We know that GRACE data over the oceans are very sensitive to the GIA and geocenter corrections (see Blazquez et al, 2018 for instance). How do the authors deal with that issue?

Why don't the authors have considered the GRACE data over Greenland and Antarctica to force the model with freshwater flux instead? As they are interested in global salinity changes, this might be more appropriate (in my point of view). I am not saying this is the wrong way to proceed. My feeling is that assimilating freshwater flux instead of ocean bottom pressure would be more appropriate to investigate the global salinity change.

Minor comments

L25-26 : The warming climate resulted in the melting of land ice but and the global ocean received net freshwater flux but not only. Ocean warming has stored more than 90% of the excess of planetary imbalance over 1970-2010 (Rhein et al., 2013; Abraham et al., 2013) in term of heat. Only 3% of the imbalance served to melt continental ice (von Schuckmann et al 2016). In term of global mean sea level, the freshwater flux accounts for 2/3 of the global mean sea level rise. Thermal expansion / ocean warming explains the other third (see WCRP Global mean sea level budget group, ESSD, 2018). This imbalance is very important and should be mentioned to increase the motivation of such a study. This is just a recommendation. I let the authors to decide whether or not they want to follow my recommendation.

L46 : "Therefore, using sea surface and near-surface salinity alone to infer the freshwater flux may incur large errors" . Can you quantify "large errors"? Can you put any estimate to these errors?

L81-84 : I acknowledge the authors to mention the ocean model they consider does not resolve eddy variability. What is the potential impact on the results? Do the authors have any idea or thought?

L113-114 : How do you remove the annual signal ?

L116: How do you obtain the dof? Can you explain the methodology me a bit more here?

L126: How is it possible to see a seasonal signal if it has been removed as previously mentioned? I am confused here. Could the authors clarify this point?

L127: Do you have any explanation for the contour plots showing wave patterns? Is it physical or not?

L131-133: I totally agree with the authors. This is a good point.

L136: Could the authors define what they call salt plume? As I understand, the salt plume is the salt

flux along the z axis. Is it correct? How do the authors estimate the salt plume? How is it diagnose (on line or off line)? Could the authors provide the equation along with a short description (in the appendix for instance) to clarify the different term considered in the salt flux?

L145 : The value are not statistically different from zero as the salt flux bring almost all the uncertainty. It is hard to conclude here. How much is the volume fraction of the ocean that is statistically different from zero? This might help for the conclusion.

L172: That is not significant though....

L176: Can you express the salt flux in freshwater anomalies for instance? This could be compared to the arctic sea ice volume change or land ice freshwater input into the oceans.

L189: It would be easier to have the same scales for the x axis on the plots to interpret and compare the results. Would it be possible ?

L443 : Why is 22 and not 23 years ? The considered period is 1993-2015 that is 23 years, right? Please, clarify throughout the paper.

Reviewer #3 (Remarks to the Author):

This manuscript takes advantage of the 1992-2015 ECCO state estimate to evaluate long-term trends in global ocean salinity and to evaluate the mechanisms driving these changes. Salinity is an invaluable tracer of freshwater input to the global ocean, and in a changing climate, it can carry fingerprints of changing evaporation and precipitation, sea ice freezing and melt patterns, and net inputs of freshwater from melting continental glaciers. This study focuses in particular on the vertical displacements of salt (or freshwater) once it enters the ocean. The results show a substantial vertical transport of salt, primarily via an advective flux (rather than diffusion). The topic is extremely pertinent, and the results show intriguing patterns that merit further exploration. However, I have a number of reservations about the particular approach adopted in this study that make me think the study might be better suited for a longer format paper, in which there would be space to fully explore the details of this analysis. In my review I summarize some of the challenges that this study presents.

(1) The study makes a credible effort (in the Appendix) to explore the fidelity of ECCO relative to the major data source from Argo. However, the Argo data cover only about half of the ECCO time period, and the major transitions in the ECCO results appear to occur around 2004, just at the juncture when Argo data become readily available. This leaves open the possibility that the long-term trends in ECCO could result from the additional constraints provided by Argo, rather than long-term shifts in oceanic salinity. This topic should be addressed with some care in the manuscript.

(2) Confusingly, although the manuscript describes ECCO as being available from 1992-2015, implying 26 years, Figures 4 and 5 reference a 22-year mean of salt fluxes. What time period was used for the calculations? Figure 3 reports changes from 1993 to 2015. Does that mean no data are used for 1992?

(3) In general, in its present succinct form, the study does not provide sufficient information to show readers how the calculations are carried out. As a reader, I felt I needed more information on the calculation of the advective and diffusive fluxes. Are all terms, including the salt plume archived in ECCO? Are they archived in such a way that the budget is guaranteed to close? Is there a residual

term in the budget? This information will be important to readers in interpreting the overall results?

(4) The manuscript is also short on general information about the ECCO run. Does ECCO have freshwater input from glacial runoff? How reliable is the sea ice model in this particular ECCO release? What is the source of E-P for ECCO, and how much are E-P values perturbed as part of the 4DVar procedure?

(5) The manuscript is also vague about depth ranges. Lines 86-87 distinguish "the surface and the upper ocean" from the "deep and abyssal ocean", but do not provide depth ranges. Those familiar with Argo will know that "upper ocean" is probably the top 2000 m, but keep in mind that the Deep Ocean Observing System (DOOS) defines the deep ocean to include everything below the euphotic zone.

(6) At lines 234-235, the authors indicate that the upper ocean "will be the only relatively well-observed part of the global ocean for a while". This doesn't really acknowledge that the global Deep Argo program is moving forward with substantial deployments in the South Indian and South Atlantic Oceans.

(7) The statistics of the analysis are not clearly presented. The authors indicate 22 degrees of freedom, presumably from 23 or 24 years of monthly data, after removal of an annual cycle. My initial thought was that the data set should provide 24x12 monthly values, and removing a sinusoidal annual cycle would require fitting a constant, an annual sine, and an annual cosine, which would remove 3 degrees of freedom, leaving 24x12-3. Clearly I don't understand how the calculation was done. I initially read this on an airplane without access to the internet to look at Liang et al (2015), and I felt that further information was needed so that the paper would stand on its own. Now that I've looked at Liang et al (2015), I'm see that it doesn't really provide a full accounting of the procedure. A revised manuscript should provide more detail about the procedure. The averages that appear in figures, such as the profile averages in Figures 2 and 4, lead to further uncertainties, as they clearly represent a spatial average. How are uncertainties determined for spatial averages?

(8) Although this may be a minor point, the paper is not formatted to conform to Nature style requirements, with numbered footnotes for references, no numbered headings, a separate Methods and Materials section, and clearly distinct Supplementary Material. Formatting issues don't impact the ability of reviewers to consider a manuscript, but they do make it hard for reviewers to consider how well the paper will convey its message to journal readers. See <https://www.nature.com/documents/ncomms-submission-guide.pdf>

(9) The study contains numerous grammatical errors---missing words, misconjugated verbs and should be proofread more carefully if possible. I provide a short list, but there may be others.

line 12: coverages  coverage

line 12: the existing  existing

line 12: the global  global

line 13: mostly focused  focused mostly

line 25: resulted  has resulted

line 52: very much limited  limited

line 73: least square  least squares

line 76: the data assimilation  data assimilation

line 78: ECCO estimate  the ECCO estimate

line 113: their annual  annual

line 131: aware of  aware

line 160: become  becomes

line 181: interfere  interfere with

line 224: on global level  on a global level

line 227: such depth  "this depth" or "such a depth"

Reviewer #1 (Remarks to the Author):

This paper is well written and uses the ECCO model to make a consistent picture of salt redistribution in the vertical over the past two decades. It makes the point that sub-mixed layer fluxes matter when interpreting surface water fluxes. I have no problem with that; it's certainly a valid point.

Thank you for the constructive comments. In this revision, we added more discussion to link our analyses to some of the major climate questions. We also made efforts to conduct some of the analyses you suggested. However, we feel the new analyses in density coordinate belong to a stand-alone paper and might distract from the major story here, which is more relevant to interpreting ocean observations and is conventionally conducted in geopotential coordinates. Also, the major results require accurate budget analyses, which are challenging to conduct in density coordinate. We thus keep the analyses conducted in the geopotential coordinate in the revised manuscript. Clarification about salt plume and more discussion about the uncertainties were also added. The detailed responses are as follows.

But so what? Having done the analysis what can you say about: 1. The mean salinity of the ocean? Is it freshening due to ice melt? It is not likely that we can yet tell but what would be needed to see that? This must depend on salt and mass conservation in the ECCO model, so a confirmation that they are indeed truly conserved is important.

Changes in the mean salinity of the global ocean largely represent the total input of freshwater, which over a sufficiently long time is likely primarily related to ice melt (e.g., glacier, ice sheet and even sea ice). Wunsch (2018) provided a concise discussion about the changes in global mean salinity from ECCO v4 and its implication on the total freshwater input estimate. Rather than focusing on the mean salinity of the global ocean, the current study is focused on better interpreting the *upper ocean* salinity changes, which are relatively well-observed through the Argo program. By examining the ECCO estimates, we can estimate the possible contribution of the vertical redistribution of salt to the changes in the global upper ocean salinity. More importantly, this study reveals the regions and dynamical processes that are important for the exchanges of the upper and deep oceans, not only for salt but also for temperature, carbon, etc. In other words, this study is useful for understanding the ocean uptakes of heat and carbon. These points were discussed in the revised manuscript (lines: 178-188).

Also, this study has an important implication for future studies on closing the ocean freshwater budget with solely observations. By combining other datasets, such as GRACE estimates of glacier and ice sheet melting, salinity changes from Argo, and sea level changes from altimeter measurements, people can potentially do a freshwater budget in a warming climate (though likely with substantial error bars) following similar approach used by Wadhams and Munk (2004). This implication for closing freshwater budget and water cycles was also added in the discussion sessions. See lines 170-177.

Both the salt and mass (volume) are conserved in the ECCO estimates. Note that since the MITgcm (ECCO model) is a Boussinesq model, mass conservation is actually presented as volume conservation. Previous studies already confirmed the conservation of salt and volume. See, e.g., Vinogradova and Ponte (2017) for formulation of salinity budget under realistic freshwater flux. In addition, Piecuch (2017) provides a detailed online description and validation of the budget conservations for volume, heat, and salt (ftp://ecco.jpl.nasa.gov/Version4/Release3/doc/v4r3_budgets_howto.pdf). In the revised manuscript, the conservations of salt and mass were explicitly mentioned (lines: 194-199). Here for your reference, plots showing the volume (mass) and salt conservations for the horizontally averaged values are as below (Figures r1, r2).

Figure r1 (volume balance, from Liang et al 2017) Global integrals of the upward (red) and downward (blue) volume transports and their sums (black). The Eulerian, eddy-induced and residual vertical transports are shown as solid, dash and dotted lines, respectively. Note the sums of the integrals of positive (negative) Eulerian and eddy-induced vertical velocities are not equal to the integrals of the positive (negative) residual vertical velocity. The sums of the upward and downward transports at any depth is zero, confirming the volume is conserved.

Figure r2: (salt balance) 22-year (Jan. 1993 – Dec. 2014) changes of the horizontally averaged salinity (blue, $\delta\langle S \rangle_{\text{total}}$), which perfectly match and thus are covered by the red curves, as well as the parts due to the net vertical salt flux (yellow, $\delta\langle S \rangle_{\text{flux}}$) and the salt plume (green, $\delta\langle S \rangle_{\text{plume}}$). The sum of the components (red, $\delta\langle S \rangle_{\text{flux}} + \delta\langle S \rangle_{\text{plume}}$) and the residuals (purple dots) are also depicted. Note that the residual is close to zero. The unit is psu year^{-1} . The upper 200 m are excluded to avoid direct impacts from the large values of each components.

Reference:

Wunsch, C. (2018). Towards determining uncertainties in global oceanic mean values of heat, salt, and surface elevation. *Tellus A: Dynamic Meteorology and Oceanography*, 70(1), 1-14.

Piecuch, C. (2017). A Note on Practical Evaluation of Budgets in ECCO Version 4 Release 3. <http://hdl.handle.net/1721.1/111094>

Liang, X., Spall, M., & Wunsch, C. (2017). Global ocean vertical velocity from a dynamically consistent ocean state estimate. *Journal of Geophysical Research: Oceans*, 122(10), 8208-8224.

Wadhams, P., and W. Munk (2004), Ocean freshening, sea level rising, sea ice melting, *Geophysical Research Letters*, 31(11), doi:Artn L1131110.1029/2004gl020039.

Some discussion of the surface boundary conditions on salinity and freshwater that are used in ECCO is appropriate. Surely the surface salt flux must be zero everywhere! (Fig. 3b). Water fluxes are non-zero due to evaporation, precipitation and ice formation but salt fluxes should always be zero.

Information about the surface boundary conditions on salinity and freshwater in ECCO estimates were added. See lines 215-227. The only salt flux at the surface is the salt exchange between ocean and sea-ice. In ECCO v4r3, the sea-ice has a salinity of 4psu. When sea-ice melts, it releases salt into the ocean. And when sea-ice forms, it will take salt out of the ocean. The net surface salt flux shown in Fig. 3b is non-zero, because there is a net melting of sea ice over the period studied.

2. What are the implications for a changing water cycle? That is, it is stated that the sub-surface flux is comparable to the surface flux, so the obvious question is how do these results affect the interpretation of Durack et al (2012) on the trends in the global water cycle. It seems that some modifications of the regional water cycle changes inferred by Durack's SSS pattern trends is called for. This should be a more tractable problem than (1) above and would show the value of the calculation.

Previous studies based on ECCO v4 (e.g., Vinogradova and Ponte, 2013; Vinogradova and Ponte, 2017) already carefully presented and discussed the implications of spatial patterns of surface salinity in inferring water cycle information. They found that the spatial patterns of the trends over the ECCO period (~20 years) are not as significant as revealed in Durack et al (2012), which examined changes over a much longer time period (~50 years). They also noted that over this period, ocean processes make it very difficult to draw simple conclusion from the spatial patterns of salinity changes. In contrast to the previous studies, this current analysis is focused on the changes of the *horizontally averaged salinity* in different layers of the global ocean and how to interpret those changes, although we briefly described some relevant spatial patterns (lines: 87-94).

Reference:

Vinogradova, N. T., and R. M. Ponte (2013), Clarifying the link between surface salinity and freshwater fluxes on monthly to interannual time scales, Journal of Geophysical Research-Oceans, 118(6), 3190-3201, doi:10.1002/jgrc.20200.

Vinogradova, N. T., and R. M. Ponte (2017), In Search of Fingerprints of the Recent Intensification of the Ocean Water Cycle, Journal of Climate, 30(14), 5513-5528, doi:10.1175/Jcli-D-16-0626.1.

In addition to these fundamental motivating questions, the basic analysis can be challenged. That is, just describing the vertical distribution of salt mixes up trends in heaving of density surfaces with vertical mixing across density surfaces. It would be nice to see this analysis re-done in density space rather than depth to properly get at the vertical mixing. It is also troubling to see such large error bars in figures 2 and 4.

Analyses in density coordinate are useful to understand the movement of water masses and can provide important insights into ocean dynamics. However, they are not particularly good for this study for at least two reasons: 1) one major advantage of ECCO v4 comparing to other products is that the mass, properties (temperature, salinity) are conserved. Since the model that ECCO used (MITgcm) is a z-coordinate configuration, the budgets of temperature and salinity are beautifully closed in the native coordinate. To conduct the analyses along density surface, we need to do interpolations, which will unavoidably introduce errors. From our experience, it is very challenging to close the budget in density

coordinate after interpolation, particularly for realistic simulations. If we focus on regional dynamical questions, that likely won't be a big problem. But for this study, which is focused on the global ocean and is wholly based on budget analyses, it is really challenging to do so. 2) A large number of previous observational studies on changes of ocean heat content and salinity were conducted in z-coordinate. For instance, IPCC AR5 present temperature and salinity changes within different depth ranges rather than density range. Using density coordinate makes it difficult to link our study to the existing literature on ocean changes. Therefore, we choose to keep the analyses in depth coordinate.

That said, analyses conducted in z-coordinate can still provide useful insights into dynamical processes. As described in the method section (lines: 228-240), the vertical salt flux includes contributions from vertical advection (residual velocity), isopycnal mixing, and diapycnal mixing. We can potentially further separate the vertical salt flux presented in the current study to components that are associated with those dynamical processes. However, since those analyses will likely distract the readers from the main information of this paper, we prefer to present those results in another paper, focused on the detailed dynamical processes that are responsible for the vertical transport of ocean properties.

Regarding the large error bars, as reviewer 3 kindly pointed out, in the previous calculation, we significantly underestimated the degrees of freedom. In the revision, we corrected the calculation, and now the error bars are significantly reduced. Discussions about the error bars were also extended. See lines 241-251.

Because of these fundamental issues, I do not recommend publication in its present form. In addition I think the term "salt plumes" is rather odd and at least needs better definition. Salt plumes from ice formation would be very small-scale (cm) so way below model resolution. Do you mean something like the Mediterranean out-flow water cascading down the slope in the Atlantic?

Here the term "salt plumes" stands for the vertical redistribution of the salt rejected during sea-ice formation. And yes, the salt plumes are not resolved but parameterized in ECCO v4. This is clarified in the revised manuscript. See lines 204-214.

I also take issue with the ECCO salinity maps that in some cases do not look much like the ARGO maps (Figures A1 and A2) . This is especially true of the mid-depth Atlantic in A2!

We appreciate that you pointed this out. Due the insufficient salinity measurements, the oceanography community currently don't have the "truth" of global ocean salinity, which is one of the motivations of the current study. It should be noted that people are inclined to believe Argo results are the "truth". However, we are conducting an intercomparison of gridded Argo products from different institutions. The preliminary results (e.g., Figure r3) reveal that various gridded Argo products, which are based on similar raw Argo measurements, actually show quite significant discrepancies. The mapping methods and assumptions in producing the gridded ocean salinity introduce various uncertainties that are difficult to quantify. It is therefore not surprising to see differences between the ECCO results and the Argo maps from Scripps. That said, the current ECCO salinity estimate indeed shows some regional features that are inconsistent with some of the other products, such as the region you mentioned, which require further analyses and improvements. In this revision, we explicitly discuss the uncertainty of the ECCO estimates as well as the results. See lines 252-263.

Figure r3: Intercomparison of the temporal means of the layered averaged salinity (0-700 m) from five Argo based gridded products (i.e., BOA, IPRC, RG, MOAA and EN4). (a) time mean (2005 through 2015) and standard deviation (stippling) of the Ensemble Mean and (b) its Ensemble Spread (i.e., standard deviation across the datasets), (c-g) difference between each individual dataset and the Ensemble Mean, and (h) largest contribution to the Ensemble Mean from the datasets (i.e., largest difference value. The range for each color is the same as the Ensemble Spread). (unit: psu). It is clear that the different gridded Argo products show various regional patterns, even for the most fundamental quantities like temporal mean.

Reviewer #2 (Remarks to the Author):

Let me first share my name with the authors: William Llovel.

The paper deals with ocean salinity change on a bidecadal basis based on the last version of the ECCO ocean reanalysis (ECCOv4r3). The authors justify the need to investigate the salinity change as it is an essential proxy for detecting and estimating the global freshwater input into the ocean. In the context of the actual global warming, mountain glaciers and ice sheets are losing mass increasing the freshwater input into the oceans. Therefore, the topic of the study is a great interest. Because of the lack of in situ measurements of salinity compare to temperature change for the last decades, salinity change and freshwater flux have been under investigated compared to temperature change.

One of the fundamental results of the study is the importance of the vertical redistribution of salt while estimating the net freshwater input. According to the authors, if we ignore the vertical redistribution of salt, the freshwater budget may be biased by 42.9% at 200m, 17.2% at 500m and less than 2% at 700m on a global basis.

This manuscript is well-written, and the results are very interesting, the conclusions reached are sufficiently important to be published for future reference. The methodology is understandable, and the authors do not over-interpret the results -which is very good-. Thus, I have no major fundamental concerns about the paper.

Thanks for the encouraging and helpful comments and suggestions. Our detailed responses to your comments and suggestions are as follows.

I do, however, have a few concerns and questions that I would like the authors to address before recommending the paper for acceptance:

- ECCO ocean reanalysis is assimilating in situ and remote sensing observations. Therefore, the reanalysis tends to match the model to the existing observations. However, is this fit done for the good reason? I am not sure this is the focus of the paper but few words might be useful to warn the reader about the skills and the limitations of such a tool.

The primary advantage of the approach ECCO used, adjusting parameters and forcings to match the estimates to observations, is that the estimates from ECCO (i.e., state variables, forcings and parameters) are dynamically consistent. In other words, all the estimates accurately follow the model equations. Therefore, we can conduct accurate budget analyses with ECCO, which is essential for quantifying the contribution of the vertical salt redistribution to the ocean salinity changes. Indeed, as you kindly pointed out, there are surely uncertainties in the estimates, largely due to the lack of enough observations and to imperfect representation of dynamical processes in the model. Nevertheless, the solution is in general consistent with the available observations within the estimated data uncertainties. The justification to ECCO v4 for the current study was added (lines: 194-199). Also, to remind the readers about these uncertainties and limitations, a separate paragraph was added in the revision (lines 241-251).

- GRACE data are assimilated in the ECCO reanalysis. But, as I understand, only GRACE data over the oceans have been considered. We know that GRACE data over the oceans are very sensitive to the GIA and geocenter corrections (see Blazquez et al, 2018 for instance). How do the authors deal with that

issue? Why don't the authors have considered the GRACE data over Greenland and Antarctica to force the model with freshwater flux instead? As they are interested in global salinity changes, this might be more appropriate (in my point of view). I am not saying this is the wrong way to proceed. My feeling is that assimilating freshwater flux instead of ocean bottom pressure would be more appropriate to investigate the global salinity change.

The GRACE fields used and respective processing, along with data weights used in the estimation, are described in Fukumori et al. (2019). The data weights account for sources of uncertainty such as those related to the geocenter and GIA corrections mentioned above. Forcing with freshwater fluxes inferred from GRACE, as suggested, is desirable but carries other uncertainties (e.g., spatial distribution around the coasts is not readily available) and has not been implemented in this solution.

Also, after obtaining the total freshwater input through better interpreting the ocean salinity measurements, we can potentially combine the GRACE data on land ice, altimeter data, as well as E-P estimates to provide a closure of the global ocean freshwater budget. This implication was added (lines: 170-177).

Reference:

Fukumori et al. (2019), Data sets used in ECCO Version 4 Release 3, <http://hdl.handle.net/1721.1/120472>

Minor comments

L25-26 : The warming climate resulted in the melting of land ice but and the global ocean received net freshwater flux but not only. Ocean warming has stored more than 90% of the excess of planetary imbalance over 1970-2010 (Rhein et al., 2013; Abraham et al., 2013) in term of heat. Only 3% of the imbalance served to melt continental ice (von Schuckmann et al 2016). In term of global mean sea level, the freshwater flux accounts for 2/3 of the global mean sea level rise. Thermal expansion / ocean warming explains the other third (see WCRP Global mean sea level budget group, ESSD, 2018). This imbalance is very important and should be mentioned to increase the motivation of such a study. This is just a recommendation. I let the authors to decide whether or not they want to follow my recommendation.

This is a very good suggestion. In the revised manuscript, we added to the introduction section the impacts of net freshwater input in the global mean sea level rise, which as you said increase the motivation. See lines 25-30.

L46 : "Therefore, using sea surface and near-surface salinity alone to infer the freshwater flux may incur large errors" . Can you quantify "large errors"? Can you put any estimate to these errors?

Estimating the likely errors is the one of the motivations of this study. After doing all the analyses we get some possible estimates of these errors. The estimates were echoed in the discussion of the paper. See lines 151-161.

L81-84 : I acknowledge the authors to mention the ocean model they consider does not resolve eddy variability. What is the potential impact on the results? Do the authors have any idea or thought?

Considering the agreements between the ECCO estimates and observations (presented here and in previous studies), we think being unable to resolve eddies won't have significant impacts on the basin or global scale results. The impacts of the unresolved eddies should be included in the adjusted parameters that were used in the eddy parametrization model (e.g., Gent-McWilliams, Redi etc.) However, the ECCO v4 products might not be a good choice if people want to focus on the dynamics of mesoscale eddies or smaller scales. In the revision, a few sentences were added to emphasize this. See lines 252-263.

L113-114 : How do you remove the annual signal?

The annual signal was removed by fitting a constant, an annual sine, and an annual cosine to the original estimates. This was made clear in the revised text (lines: 241-243).

L116: How do you obtain the dof? Can you explain the methodology me a bit more here?

As reviewer 3 pointed out, the dof in the original manuscript was underestimated. In this revision, the dof is calculated as follows. ECCO v4 r3 have 288 monthly-averaged time records (Jan 1992 - Dec 2015). Because of the quality of the estimates, the first and last years are not included, which leaves 264 data points. After removing the annual signal, 3 dof are further reduced, leaving the total dof at 261. This is added in the method section. See lines 244-246.

L126: How is it possible to see a seasonal signal if it has been removed as previously mentioned? I am confused here. Could the authors clarify this point?

The high-frequency "signal" is the residual after removing the fitted annual signals. Note on the high frequency signals was added in the caption of Figure 1.

L127: Do you have any explanation for the contour plots showing wave patterns? Is it physical or not?

It is the small residual left after removing the fitted annual cycle. Note is added in the caption of Fig1.

L131-133: I totally agree with the authors. This is a good point.

Thanks!

L136: Could the authors define what they call salt plume? As I understand, the salt plume is the salt flux along the z axis. Is it correct? How do the authors estimate the salt plume? How is it diagnose (on line or off line)? Could the authors provide the equation along with a short description (in the appendix for instance) to clarify the different term considered in the salt flux?

Here the term "salt plumes" stands for the vertical redistribution of the salt rejected from sea-water during sea-ice formation. The salt plumes were presented in the tracer equation as a source/sink term on the right rather than a flux term. In ECCO v4r3, the plumes are not resolved but parameterized following previous literature. This is clarified in the revised manuscript. See lines 206-212.

L145 : The value are not statistically different from zero as the salt flux bring almost all the uncertainty. It is hard to conclude here. How much is the volume fraction of the ocean that is statistically different from zero? This might help for the conclusion.

Regarding the large error bars, as reviewer 3 kindly pointed out, in the previous calculation, we significantly underestimated the degrees of freedom. In the revision, we corrected the calculation, and now the error bars are significantly reduced and most of the fluxes are significantly from zero, particularly for the advective and diffusive components.

L172: That is not significant though....

Please refer to our response to the comment just above. Also, please note that in the revision we make sure that all the calculation done in a 22 years period (Jan 1993-Dec 2014), so there are very small changes in the values of the vertical salt fluxes.

L176: Can you express the salt flux in freshwater anomalies for instance? This could be compared to the arctic sea ice volume change or land ice freshwater input into the oceans.

The salt fluxes were converted to freshwater anomalies. The comparison of the results to land ice freshwater input to the oceans were added. See lines 114-116 and 155-159.

L189: It would be easier to have the same scales for the x axis on the plots to interpret and compare the results. Would it be possible?

Plots were adjusted following your suggestion.

L443 : Why is 22 and not 23 years ? The considered period is 1993-2015 that is 23 years, right? Please, clarify throughout the paper.

Here 22 years means the period from Jan 1993 through Dec 2014. To avoid misunderstanding, related statements throughout the manuscript are revised.

Reviewer #3 (Remarks to the Author):

This manuscript takes advantage of the 1992-2015 ECCO state estimate to evaluate long-term trends in global ocean salinity and to evaluate the mechanisms driving these changes. Salinity is an invaluable tracer of freshwater input to the global ocean, and in a changing climate, it can carry fingerprints of changing evaporation and precipitation, sea ice freezing and melt patterns, and net inputs of freshwater from melting continental glaciers. This study focuses in particular on the vertical displacements of salt (or freshwater) once it enters the ocean. The results show a substantial vertical transport of salt, primarily via an advective flux (rather than diffusion). The topic is extremely pertinent, and the results show intriguing patterns that merit further exploration. However, I have a number of reservations about the particular approach adopted in this study that make me think the study might be better suited for a longer format paper, in which there would be space to fully explore the details of this analysis. In my review I summarize some of the challenges that this study presents.

Thanks for the constructive comments and suggestions. In this revision, we significantly expanded the method section and provide readers concise information about the ECCO products, particularly how the salinity budget was closed, what the surface boundary conditions are, how the salt flux was calculated, and the associated uncertainties. We also tried to clarify the text to make it more readable. Our detailed responses to your comments are as follows.

(1) The study makes a credible effort (in the Appendix) to explore the fidelity of ECCO relative to the major data source from Argo. However, the Argo data cover only about half of the ECCO time period, and the major transitions in the ECCO results appear to occur around 2004, just at the juncture when Argo data become readily available. This leaves open the possibility that the long-term trends in ECCO could result from the additional constraints provided by Argo, rather than long-term shifts in oceanic salinity. This topic should be addressed with some care in the manuscript.

Since Figure 1 shows only the salinity *anomaly*, the likely reason for the change of sign occurring around 2004 is that year is about the middle of the examined period. That said, the reviewer's question on the impacts of assimilating Argo is important. Two of the authors, Liang and Liu, are actually evaluating a number of ocean reanalyses products including ECCO v4, and examining the impacts of assimilating different data sets, particularly Argo. What we did is to calculate and examine the time series of the global halo-steric height, which is not expected to change dramatically before and after assimilating Argo data. Preliminary results show that many of the examined ocean reanalysis products do introduce spurious changes after starting to assimilate Argo data, however, ECCO v4 does not show such an impact (Figure r4). These results, in addition to the good agreements between ECCO and Argo as well as other previous studies described in the supplementary document, gave us much confidence about the trends.

Another related issue is possible model drifts. By conducting runs with the ECCO v4 model to simulate the dispersal of biogeochemical tracers, Forget et al (2015) shows that the adjusted mixing parameters utilized in ECCO v4 significantly reduced spurious model drifts. Considering all these, we think ECCO v4 does a reasonable job in presenting ocean salinity changes with what we have today. We admit that the ultimate way to confirm the estimate is to get more long-term salinity measurements, which might take ages to occur. In the revised manuscript, we added more discussion about the uncertainties and possible problems of the ECCO v4 estimates. See lines 252-263.

Reference:

Forget, G., D. Ferreira, and X. Liang (2015b), On the observability of turbulent transport rates by Argo: supporting evidence from an inversion experiment, *Ocean Science*, 11(5), 839-853, doi:10.5194/os-11-839-2015.

Figure r4: Time series of the horizontally and depth-averaged steric height anomaly: thermo-steric and halo-steric height of ECCOv4r3. (unit: m). No significant impacts of assimilating Argo were revealed.

(2) Confusingly, although the manuscript describes ECCO as being available from 1992-2015, implying 26 years, Figures 4 and 5 reference a 22-year mean of salt fluxes. What time period was used for the calculations? Figure 3 reports changes from 1993 to 2015. Does that mean no data are used for 1992?

Sorry for the confusion. ECCO v4r3 is available from Jan 1992 to Dec 2015, so there is a total of 24 years of data. Due to the quality of estimates, the first and last years were not used in this study. Therefore, only 22 years data (Jan 1993 – Dec 2014) were utilized. In this revision, all the related statements were revised accordingly.

(3) In general, in its present succinct form, the study does not provide sufficient information to show readers how the calculations are carried out. As a reader, I felt I needed more information on the calculation of the advective and diffusive fluxes. Are all terms, including the salt plume archived in ECCO? Are they archived in such a way that the budget is guaranteed to close? Is there a residual term in the budget? This information will be important to readers in interpreting the overall results?

To address these comments, we significantly expanded the method section. More specifically, we added 1) details about the boundary conditions, particularly the surface freshwater and salt fluxes (lines: 215-227); 2) more information about “salt plume” (lines: 206-212); 3) references showing how the salt budget is closed and confirmed (lines 194-199); 4) brief description on how the vertical advective and diffusive fluxes were calculated (lines: 228-240). The much expanded method section can be found in lines 190-263.

In particular, both the salt and mass (volume) are conserved in the ECCO estimates. Note that since the MITgcm (ECCO model) is a Boussinesq model, mass conservation is actually presented as volume conservation. Previous studies already confirmed the conservation of salt and volume. See, e.g., Vinogradova and Ponte (2017) for formulation of salinity budget under realistic freshwater flux. In addition, Piecuch (2017) provides a detailed online description and validation of the budget conservations for volume, heat, and salt (ftp://ecco.jpl.nasa.gov/Version4/Release3/doc/v4r3_budgets_howto.pdf). In the revised manuscript, the conservations of salt and mass were explicitly mentioned (lines: 194-199). Here for your reference, plots showing the volume (mass) and salt conservations for the horizontally averaged values are as below (Figures r5, r6).

(4) The manuscript is also short on general information about the ECCO run. Does ECCO have freshwater input from glacial runoff? How reliable is the sea ice model in this particular ECCO release? What is the source of E-P for ECCO, and how much are E-P values perturbed as part of the 4DVar procedure?

Briefly, there is no special treatment of glacial runoff. Previous evaluation shows that the sea ice in the estimate is reasonable. E-P in ECCO comes from ERA interim and at least the global net input of freshwater is strongly adjusted to account for imbalances in original fields. For details of these information, refer to Forget et al. (2015a). A brief description of those information was also added in the expanded method section. See lines 215-227.

Reference:

Forget, G., J. M. Campin, P. Heimbach, C. N. Hill, R. M. Ponte, and C. Wunsch (2015a), ECCO version 4: an integrated framework for non-linear inverse modeling and global ocean state estimation, Geoscientific Model Development, 8(10), 3071-3104, doi:10.5194/gmd-8-3071-2015.

(5) The manuscript is also vague about depth ranges. Lines 86-87 distinguish "the surface and the upper ocean" from the "deep and abyssal ocean", but do not provide depth ranges. Those familiar with Argo will know that "upper ocean" is probably the top 2000 m, but keep in mind that the Deep Ocean Observing System (DOOS) defines the deep ocean to include everything below the euphotic zone.

Thanks for pointing this out. In the revision, we clearly state that the upper ocean in this paper means the layer above 2000 m. See lines 57-60.

(6) At lines 234-235, the authors indicate that the upper ocean "will be the only relatively well-observed part of the global ocean for a while". This doesn't really acknowledge that the global Deep Argo program is moving forward with substantial deployments in the South Indian and South Atlantic Oceans.

The global Deep Argo program was acknowledged in the revised manuscript and was also discussed in the context of reducing the existing uncertainties of ECCO estimates. See lines 185-188.

(7) The statistics of the analysis are not clearly presented. The authors indicate 22 degrees of freedom, presumably from 23 or 24 years of monthly data, after removal of an annual cycle. My initial thought was that the data set should provide 24x12 monthly values, and removing a sinusoidal annual cycle would require fitting a constant, an annual sine, and an annual cosine, which would remove 3 degrees of freedom, leaving 24x12-3. Clearly I don't understand how the calculation was done. I initially read this on an airplane without access to the internet to look at Liang et al (2015), and I felt that further information was needed so that the paper would stand on its own. Now that I've looked at Liang et al (2015), I'm see that it doesn't really provide a full accounting of the procedure. A revised manuscript should provide more detail about the procedure. The averages that appear in figures, such as the profile averages in Figures 2 and 4, lead to further uncertainties, as they clearly represent a spatial average. How are uncertainties determined for spatial averages?

Thanks for the useful suggestion. The calculation on dof and error bars are revised accordingly. Now the dof is $22 \times 12 - 3$, 261 (22-year for 1993-2014). Note that the errors in figs 2 and 4 are based on the time variability (standard deviation) of the spatially averaged time series. And the spatial variation was not taken into account in calculating the error bars. This information was made clear in the method section (lines 241-251).

(8) Although this may be a minor point, the paper is not formatted to conform to Nature style requirements, with numbered footnotes for references, no numbered headings, a separate Methods and Materials section, and clearly distinct Supplementary Material. Formatting issues don't impact the ability of reviewers to consider a manuscript, but they do make it hard for reviewers to consider how well the paper will convey its message to journal readers. See <https://www.nature.com/documents/ncomms-submission-guide.pdf>

The revised manuscript is now formatted following the Nature style requirements.

(9) The study contains numerous grammatical errors---missing words, misconjugated verbs and should be proofread more carefully if possible. I provide a short list, but there may be others.

line 12: coverages  coverage
line 12: the existing  existing
line 12: the global  global
line 13: mostly focused  focused mostly
line 25: resulted  has resulted
line 52: very much limited  limited
line 73: least square  least squares
line 76: the data assimilation  data assimilation
line 78: ECCO estimate  the ECCO estimate
line 113: their annual  annual
line 131: aware of  aware
line 160: become  becomes
line 181: interfere  interfere with
line 224: on global level  on a global level
line 227: such depth  "this depth" or "such a depth"

Grammatical errors and other language issues were corrected.

Reviewers' comments:

Reviewer #1 (Remarks to the Author):

I do not find all that much change in the revised manuscript and do not find it compelling for publication in Nature Communications. Parts of the paper read like a justification for ECCO and what could be done in future. I still feel that this exercise lacks consequence; they have not answered my "So what?" question. And I still greatly object to the use of the term "salt plumes" to refer to the way the model treats brine rejection in sea-ice formation. Salt plumes are tiny, and the authors are discussing a completely different macroscopic vertical mixing phenomena. And not describing what they mean very well at all.

I tend to agree with the 3rd reviewer that this work would be better suited for a regular journal where a fuller treatment can be given to the spatial patterns of the vertical distribution of salt and the difference between vertical heaving and changes on isopycnals could be examined. And the suspicious changes in vertical salt distribution centered around 2004 could be more carefully defended (Figure 1).

Reviewer #2 (Remarks to the Author):

Review of "Vertical Redistribution of the Global Oceanic Salt Content" by Xinfeng Liang et al.

I have read the revised manuscript by Xingeng Liang et al and the responses to reviewers' comments. The authors have answered all my previous concerns and questions in the revised manuscript.

The manuscript is well-written, and the results are very interesting. I find the conclusions reached and sufficiently important to be published for future reference. The authors do not over-interpret the results -which is very good-. I do not have any fundamental concerns about this paper.

Thus, I recommend the paper for publication in Nature Communication.

William Llovel

Reviewer #3 (Remarks to the Author):

In the revised manuscript, the authors have clarified their discussion of salt redistribution in the ECCO2 state estimate. The authors have addressed almost all of my reviewer comments in their updated manuscript with a couple of exceptions.

In my first review point (1), I noted that Figure 1 shows a major transition in salt anomalies about the time that Argo observations started, and I noted that readers might be skeptical about the conclusions if they thought that this transition was an artifact of the rapid increase in observations provided by Argo. The authors wrote a detailed response, indicating that they have been exploring this issue in a separate study, and that they are confident that the transition is a real signal. However, they seem to have assumed that no one other than reviewer 3 would really care about this issue, and as far as I can tell, they have not mentioned this issue in the revised manuscript. The authors should update the manuscript to explain their thinking. (It would also be useful to show misfits to hydrographic data and float data as a function of time to help substantiate this point.

My first review point (4) asked about the treatment of glacial runoff in ECCO. The authors have

explained in their response to reviewers that ECCO contains no glacial runoff, but their revised discussion of the freshwater budget in lines 215-227 does not mention the omission of glacial runoff. This is particularly relevant, as a number of recent studies (notably Bronselaer et al, Nature, 2018) have highlighted the role that meltwater can play in the climate system. The authors should not be afraid to identify the potential omissions of the model.

Reviewers' comments:

Reviewer #1 (Remarks to the Author):

I do not find all that much change in the revised manuscript and do not find it compelling for publication in Nature Communications. Parts of the paper read like a justification for ECCO and what could be done in future. I still feel that this exercise lacks consequence; they have not answered my “So what?” question.

The main messages of the paper (calling attention to the importance of vertical salt redistribution in the ocean, quantitatively describing advective/diffusive contributions to interior salt transports, pointing out implications for upper ocean budgets and inferences about surface freshwater fluxes from salinity measurements, among others) are clearly exposed in the text and of sufficient interest to merit publication, in our opinion.

And I still greatly object to the use of the term “salt plumes” to refer to the way the model treats brine rejection in sea-ice formation. Salt plumes are tiny, and the authors are discussing a completely different macroscopic vertical mixing phenomena. And not describing what they mean very well at all.

We replaced throughout the text the term “salt plumes” with “brine rejection.” Also, we explicitly reminded the readers that the “brine rejection” term was parameterized rather than explicitly simulated in ECCO v4r3. See lines 79-87, 223-229.

I tend to agree with the 3rd reviewer that this work would be better suited for a regular journal where a fuller treatment can be given to the spatial patterns of the vertical distribution of salt and the difference between vertical heaving and changes on isopycnals could be examined. And the suspicious changes in vertical salt distribution centered around 2004 could be more carefully defended (Figure 1).

In the last round of revision, we explained in detail why we did not think the suggested analyses, examining “the difference between vertical heaving and changes on isopycnals,” was that useful for this study. We agree that the suggested analyses can be done regionally and can provide insights into ocean dynamics. However, they belong to a separate study with other focus rather than the current one.

For the “suspicious changes” centered around the year 2004, see response to reviewer 3 below.

Reviewer #2 (Remarks to the Author):

Review of “Vertical Redistribution of the Global Oceanic Salt Content” by Xinfeng Liang et al.

I have read the revised manuscript by Xingeng Liang et al and the responses to reviewers’ comments. The authors have answered all my previous concerns and questions in the revised manuscript.

The manuscript is well-written, and the results are very interesting. I find the conclusions reached and sufficiently important to be published for future reference. The authors do not over-interpret the results -which is very good-. I do not have any fundamental concerns about this paper.

Thus, I recommend the paper for publication in Nature Communication.

William Llovel

We thank Dr. Llovel for his help in improving this manuscript.

Reviewer #3 (Remarks to the Author):

In the revised manuscript, the authors have clarified their discussion of salt redistribution in the ECCO2 state estimate. The authors have addressed almost all of my reviewer comments in their updated manuscript with a couple of exceptions.

In my first review point (1), I noted that Figure 1 shows a major transition in salt anomalies about the time that Argo observations started, and I noted that readers might be skeptical about the conclusions if they thought that this transition was an artifact of the rapid increase in observations provided by Argo. The authors wrote a detailed response, indicating that they have been exploring this issue in a separate study, and that they are confident that the transition is a real signal. However, they seem to have assumed that no one other than reviewer 3 would really care about this issue, and as far as I can tell, they have not mentioned this issue in the revised manuscript. The authors should update the manuscript to explain their thinking. (It would also be useful to show misfits to hydrographic data and float data as a function of time to help substantiate this point.

We thank the reviewer for reminding us about this. The change in sign is not a “major transition” but simply results from the presence of a linear trend. The trend naturally yields a change of sign in salinity anomaly around the middle of the period examined, which is roughly coincident with the start of the Argo era. In this revision, the change of sign of the salinity anomaly was noted in the main text. See lines 73-78.

My first review point (4) asked about the treatment of glacial runoff in ECCO. The authors have explained in their response to reviewers that ECCO contains no glacial runoff, but their revised discussion of the freshwater budget in lines 215-227 does not mention the omission of glacial runoff. This is particularly relevant, as a number of recent studies (notably Bronselaer et al, Nature, 2018) have highlighted the role that meltwater can play in the climate system. The authors should not be afraid to identify the potential omissions of the model.

ECCO v4r3 did not explicitly prescribe the glacial runoff. Now, this is stated clearly in the main text. Also, since the ECCO solution and in particular its surface freshwater fluxes are adjusted by fitting the simulation to ocean observations, the final results, including the adjusted surface freshwater input, may include the impacts of glacial runoff. This possibility is included in the revised manuscript. See lines 236-241.